# Epitaxial Growth of Ordered In-Plane Si and Ge Nanowires on Si (001)

**DOI:** 10.3390/nano11030788

**Published:** 2021-03-19

**Authors:** Jian-Huan Wang, Ting Wang, Jian-Jun Zhang

**Affiliations:** 1Beijing National Laboratory for Condensed Matter Physics and Institute of Physics, Chinese Academy of Sciences, Beijing 100190, China; jhwang1@iphy.ac.cn (J.-H.W.); wangting@iphy.ac.cn (T.W.); 2School of Physical Sciences, University of Chinese Academy of Sciences, Beijing 100190, China; 3Songshan Lake Materials Laboratory, Dongguan 523808, China

**Keywords:** in-plane nanowire, site-controlled, epitaxial growth, silicon, germanium, nanowire-based quantum devices

## Abstract

Controllable growth of wafer-scale in-plane nanowires (NWs) is a prerequisite for achieving addressable and scalable NW-based quantum devices. Here, by introducing molecular beam epitaxy on patterned Si structures, we demonstrate the wafer-scale epitaxial growth of site-controlled in-plane Si, SiGe, and Ge/Si core/shell NW arrays on Si (001) substrate. The epitaxially grown Si, SiGe, and Ge/Si core/shell NW are highly homogeneous with well-defined facets. Suspended Si NWs with four {111} facets and a side width of about 25 nm are observed. Characterizations including high resolution transmission electron microscopy (HRTEM) confirm the high quality of these epitaxial NWs.

## 1. Introduction

Si and Ge nanowires (NWs) have potential applications for high-performance transistors [1,2] and for disruptively quantum computation technology [3,4,5,6]. The controllable growth of NW arrays in wafer-scale remains the major challenge for large scale integration. The top-down method by patterning and etching can precisely fabricate NWs in wafer-scale but also induce additional defects during the nanofabrications. For instance, IMEC has previously reported the vertically stacked horizontal Si NWs with selective etching of Si/SiGe multilayer fin structures [1]. Moreover, by selectively etching Si, stacked SiGe NWs were obtained to improve the channel mobility [7]. However, the top-down fabrication introduces atomic surface roughness and damages, which deteriorate the carrier mobility of the NWs [8].

Alternatively, the self-assembled growth of NWs via a vapor-liquid-solid (VLS) mechanism can form high quality NWs with a sharp interface [9,10]. A mobility of 730 cm^2^(Vs)^−1^ [11] and a ballistic conduction up to several hundred nanometers [12] were reported in such {111}-oriented Ge/Si core/shell NWs. Compared to the {111}-oriented NWs, {110}-oriented Ge/Si core/shell NWs have substantially enhanced hole mobility as high as 4200 cm^2^(Vs)^−1^ at 4 K [13]. Although, the VLS-grown Si and Ge NWs have recently presented single crystalline with controllable orientation [14,15,16,17,18], the out-of-planar geometry has not been compatible with the well-established planar device processing technology. Ex-situ assembly methods such as contact printing and capillary assembly have been developed to align the NWs on a target substrate [19,20], however, for such VLS-grown NWs, the precise positioning at a large scale is a challenge. Another challenge of the VLS-grown NWs is the poor size-controllability (including both length and diameter), which is considered to reduce the collective properties of NWs.

Combining top-down nanofabrication and bottom-up self-assembly, we have recently demonstrated site-controlled growth of Ge hut wires on trench-patterned Si (001) substrate [21]. The Ge hut wires have a height of 3.8 nm with sharp {105} facets specifically oriented along <100> directions with high scalability. They are grown under a relatively high growth temperature where Si and Ge intermixing leads to a reduced Ge composition in the wires. Therefore, it is desirable to obtain epitaxial Si and Ge NWs with controllable size, orientation, and composition. In this work, we epitaxially grow {110}-orientated in-plane Si, SiGe, and Ge NWs on pre-patterned Si NW arrays. The pre-patterned Si NWs with an inverted trapezoidal structure are obtained through nanofabrications. On such pre-patterned Si NWs, homogeneous Si NWs with controllable sizes are epitaxially grown by molecular beam epitaxy. Furthermore, we demonstrate the formation of the conformal SiGe NWs and Ge NWs with {113} facets on the diamond-shaped Si NWs with {111} facets and truncated Si NWs. By transmission electron microscopy (TEM) characterizations, we investigate the material properties of the NWs mentioned above, which exhibit a high quality.

## 2. Materials and Methods

A CMOS-compatible top-down method was explored here to define the planar Si NW arrays on 200 mm Si (001) wafers. Figure 1 describes the fabrication process: a SiO_2_ grating structure is firstly prepared along <110> direction on Si wafer by plasma enhanced vapor deposition, deep ultraviolet lithography, and reactive ion etching. Such SiO_2_ grating structure is used as a hard mask for the subsequent wet etching of Si. The SiO_2_ grating structure has periods that range from 360 to 440 nm with a constant duty cycle of nearly 1:1 and a depth of 150 nm. After dipping for 5 s in a buffered HF solution (7:1) to remove the native oxide on the exposed Si, a diluted tetramethylammonium hydroxide aqueous solution (TMAH 5%) is used to create the planar Si NWs at 75 °C. The SiO_2_ hard mask is finally removed in diluted HF solution.

By obtaining these pre-patterned Si NWs, we then studied the direct epitaxial growth of Si NWs, SiGe NWs, and Ge/Si core/shell NWs inside a SiGe molecular beam epitaxy system (Octoplus 500 EBV, MBE-Komponenten, Weil der Stadt, Germany). The patterned wafer was cleaved into 10 × 10 mm^2^ small samples before dipping in a diluted HF solution for deoxidation and hydrogen passivation. To reduce the thermal instability of these tiny pre-patterned NWs, a low-temperature dehydrogenation was performed at 500 °C. The Si epitaxial NWs were obtained after homoepitaxial growth of Si at growth temperatures from 380 °C to 480 °C with a growth rate of 1 Å/s.

The SiGe NWs and Ge NWs were grown on the Si epitaxial NW after deposition 20 nm Si layer at 450 °C and 380 °C, respectively. The SiGe NWs were obtained by depositing 10 nm Si_0.5_Ge_0.5_ and 10 nm Si at 350 °C, where the growth rate of Si and Ge was 0.5 Å/s. The Ge NWs were obtained by depositing 2 nm Ge at 300 °C with a growth rate of 0.3 Å/s. The Ge/Si core/shell NWs were further formed after the deposition of 3 nm Si capping layer at 300 °C.

Focus ion-beam (FIB) system (NanoLab Helios 600i, FEI, Hillsboro, USA) equipped with high-resolution field-emission scanning electron microscope (SEM) was employed to elucidate the morphology of NWs and prepare the TEM lamellae. Before the FIB-milling, the NW sample was coated with 5 nm Ti and 50 nm Au for protection. TEM was performed to verify the quality of these epitaxial NWs, using a JEOL 2100 plus, operating at 200 kV.

## 3. Results and Discussion

### 3.1. Planar Trapezoidal Si NW Arrays

TMAH solution provides anisotropic wet etching for Si, with selectivity more than 1:10 between the Si {111} and Si {100} planes [22]. Therefore, {111}-faceted V-grooves were fabricated along the <110> direction, as shown in the SEM images (Figure 2a,b). In Figure 2a, on the tips of the Si V-grooves, we observed a Si hourglass figure with inverted {111} facets contributing to the SiO_2_ hard mask. With optimized etching conditions, the formation of Si NWs with an inverted triangular or trapezoidal shape are achieved. Multiple widths of Si NWs ranging from 20 nm to 40 nm can be fabricated simultaneously on 200 mm Si (001) wafer by varying the pattern sizes. Figure 2a shows trapezoidal Si NWs with a minimum width of approximately 20 nm, while still preserving good uniformity, as confirmed by the surface SEM images, as shown in Figure 2b. The average width of the neck is approximately 3 nm, as shown in the inset of Figure 2a, expected to be facilely isolated by thermal oxidation [23,24]. The lengths of NW arrays are defined ranging from 2 μm up to 2 mm, suggesting a large aspect ratio (length: width) of nearly 10^5^.

### 3.2. Homoepitaxy of Si NWs

Figure 3a presents a typical NW array by homoepitaxially grown Si on pre-patterned trapezoidal Si NWs. They are highly uniform. The width of these epitaxial Si NWs can be tuned from 30 nm to 50 nm by simply changing the growth conditions. Figure 3b shows the cross-sectional SEM image of epitaxial NWs obtained after the deposition of 20 nm Si layer on 30 nm wide pre-patterned Si NWs at 380 °C. Although only 20 nm Si were deposited at 380 °C, the epitaxial NW evolved rapidly toward the {111}-faceted morphology and a small Si (001) terrace with a width less than 10 nm on the top was left, driven by the reduction of surface energy. We observe a truncated {111}-faceted Si NW with a Si (001) terrace on the top (Figure 3b). By depositing 20 nm Si at 380 °C on a 40 nm wide pre-patterned NW array, a Si (001) terrace with enlarged width of approximately 17 nm was obtained (Figure 3c). If we increase the growth temperature to 450 °C, the Si (001) terrace will evolve into two symmetric Si (111) facets (Figure 4a), which leads to a 33 nm wide diamond-shaped NW. By keeping the growth temperature at 450 °C, when the Si layer is increased to 50 nm, the average width of diamond-shape Si NWs enlarges to approximately 48 nm (Figure 4b). The NWs are characterized by high-resolution TEMs (HRTEMs). Figure 4c provides a cross-sectional HRTEM image of a single Si NW obtained at the identical growth conditions to those in Figure 4b. The green dashed line in Figure 4c represents the interface between the epitaxial layer and the initial hourglass structure (pre-patterned trapezoidal Si NW). The inset of Figure 4c provides a zoom-in HRTEM image of the epitaxial interface labeled in Figure 4c, showing a perfect arrangement of Si atoms. Atoms deposited on the Si hourglass structure diffuse upwards to the shoulder areas to reduce the surface area, as illustrated by black arrows.

Although the pre-patterned trapezoidal NWs are thermally stable at the aforementioned low-temperature epitaxy, we note that a high-temperature dehydrogenation process at more than 600 °C will deform the pre-patterned Si NWs. The thermal instability becomes remarkable for Si NWs with smaller dimensions [25,26], as we find that the pre-patterned NWs with a size of about 20 nm in Figure 2a deform into discrete Si beads only after 500 °C dehydrogenation. Similar phenomena have been previously reported on an isolated Si NW as Plateau–Rayleigh instability (PRI) [27,28], while the critical temperature reported is much higher at 775 °C for a Si NW with 100 nm diameter. In our case, the root causes of thermal instability are not just dominated by PRI, also strongly influenced by the fragile narrow Si necks as well as the surface diffusion between NWs and patterned V-grooves.

The supporting Si neck of the hourglass structure can significantly affect the thermal instability of the NW growth with small dimensions. Here, we then study the possibility of creating suspended NWs. Figure 5a shows a typical 2 μm long suspended Si trapezoidal NW with a sub-20 nm average width. The supporting Si necks are removed by similar fabrication method mentioned above with over-etched conditions. Absence of the neck, such suspended structure can avoid the diffusion between the NW and the V-groove more effectively. After the growth of the 20 nm Si layer, although the gap between the NW and the pre-patterned V-groove appears to be unclear in the SEM picture (Figure 5b), TEM characterization in Figure 5c has verified that still retains the suspended configuration and forms {111}-faceted diamond-shaped NW with a side width of about 25 nm. Overall, the suspended Si NW exhibit enhanced thermal stability and homogeneity with four {111} facets at small dimensions, which can be considered as an ideal isolated one-dimensional NW system. But these suspended Si NWs are limited to a few micrometers in length, due to insufficient mechanical strength.

The size distributions of both the pre-patterned trapezoidal NW and epitaxial NWs were investigated. Figure 6a–c presents the SEM images of the pre-patterned NW and the epitaxial NWs obtained after the deposition of 20 and 50 nm-thick Si, respectively. After epitaxial growth, the rough surface of the pre-patterned NW has been significantly modified by forming atomic {111} facets. As illustrated in Figure 6d, the average width of pre-patterned trapezoidal NWs is 29.8 nm with relative standard deviation of 6.4%. By depositing a 20 nm (50 nm)-thick Si layer, the epitaxially formed Si NWs exhibit average widths of 35.8 nm (46.0 nm), and the relative standard deviation of the width distribution is reduced to 3.9% (2.9%).

### 3.3. Epitaxy of SiGe NWs

The epitaxial Si NWs provide platform for the subsequent growth of SiGe and Ge NWs. As mentioned, the SiGe NWs are obtained after the deposition of 10 nm Si_0.5_Ge_0.5_ and 10 nm Si layer at 350 °C on the {111}-faceted Si NW. We should note that all the thicknesses of the epitaxial layer mentioned in this work are referred to as-grown layer thickness on flat substrate. Here, the actual Si_0.5_Ge_0.5_ thickness that was deposited on the {111} facets should be 5.77 nm. The SEM images in cross-sectional view (Figure 7a,b) and top view (Figure 7c,d) indicate that these SiGe NWs are highly uniform. Attributed to the high Ge content in the SiGe layer, we can directly distinguish the SiGe layer in the magnified SEM image as shown in Figure 7b, where the SiGe layer has a brighter contrast.

Due to 2.1% lattice mismatch between Si_0.5_Ge_0.5_ and Si, misfit dislocations will generate if the SiGe film reaches the critical thickness for pseudomorphic growth. From the magnified planar SEM image (Figure 7d), the red arrow indicates strain-induced defects generated at the Si V-groove, indicating the excessive deposition of the SiGe layer. Figure 7e is a cross-sectional TEM image at the Si V-groove, showing that stacking faults (SFs) have generated from the interface and penetrated to the surface along the {111} gliding plane. In addition, we also observed other types of defects including SFs in parallel to the side-wall, attributed to plastic relaxation [29].

The situation is different for the SiGe NW. Figure 7f is a HRTEM image of directly grown in-plane SiGe/Si NW, with absence of defects, indicating the high crystal quality and conformal growth of the SiGe NW. The inset in Figure 7f is the fast Fourier transform (FFT) pattern of the SiGe/Si NW, showing only a single set of diffraction spots without distinct splitting. The FFT pattern is in-line with the spatial measurement result, indicating the SiGe NW is fully strained on Si NW.

### 3.4. Epitaxy of Ge/Si Core/Shell NWs

Despite a 4.2% lattice-mismatch between Ge and Si, we have further demonstrated Ge NW growth on the truncated {111}-faceted Si NW, where the average width of the Si (001) terrace is about 17 nm. As mentioned, the Ge NWs are obtained after the deposition of 2 nm Ge with a growth rate of 0.3 Å/s. In order to suppress the intermixing between Ge and Si, the growth is performed at a relatively low temperature of 300 °C [30]. Following a 3 nm Si capping layer deposited at 300 °C, Ge/Si core/shell NW is obtained, which can provide a high-performance one-dimensional hole gas system for exploring hole spin qubits [3,4,21]. Figure 8a,b shows cross-sectional and top view SEM images of the Ge/Si core/shell NW arrays, presenting a uniform morphology and smooth surface of NWs. To note, there are also numbers of strain-induced Ge islands formed on the Si V-grooves.

HRTEM micrograph in Figure 8c shows a typical cross-section of the Ge/Si core/shell NW. The Ge NW is grown on the <110>-oriented Si (001) terrace of the truncated {111}-faceted Si NW. The zoom-in HRTEM image in the inset of Figure 8c presents a trapezoidal geometry of the Ge NW composed of two (113) side facets and a flat (001) top surface. The formation of Ge (113) facets is attributed to the low surface energy, which has been reported in previous works [31,32,33]. Compared with <100>-oriented Ge hut wires [21,34], these <110>-oriented Ge NWs exhibit a larger aspect ratio of more than 0.2, where the height and width of the Ge NW are about 4 nm and 18 nm, respectively. Comparing the height of the Ge NW on the Si (001) terrace h_001_ ≈ 39.4 Å and the thickness of the Ge wetting layer on (111) side facets h_111_ ≈ 6.3 Å, we conclude that there is a significant Ge diffusion from the (111) facet towards the (001) facet. In terms of thermodynamics, Si (001) features higher surface energy than Si (111) [35,36], thus such Ge diffusion toward (001) facet is energetically favored.

Considering the low growth temperature, the intermixing of Ge and Si is strongly suppressed, thus we can expect an almost pure Ge-core in such Ge/Si NWs. Furthermore, atomically sharp interfaces between the Ge-core and the Si-shell are observed in the inset of Figure 8c, which further confirms the negligible intermixing between Ge and Si.

## 4. Conclusions and Perspectives

In summary, homogenous planar diamond-shaped Si NW arrays (30–50 nm in width) have been achieved on pre-patterned {111}-faceted Si arrays via direct epitaxial growth. Morphologies and dimensions of these NWs are controllable, while they can also be tuned under certain growth conditions. Suspended Si NWs exhibit diamond-shaped cross-section with four Si {111} facets. Furthermore, the SiGe NWs can be conformally grown on the {111}-faceted Si NWs. Additionally, {113}-faceted Ge NWs along [110] direction are also obtained after the deposition of 2 nm Ge on the truncated Si NWs. HRTEMs reveal the high quality of these epitaxial NWs.

The in-plane and site-controllable epitaxial NWs hold promise as the platform for the next generation of devices that require addressability and scalability. The Si and SiGe NWs have potential applications for high-perform transistors [7,23]. Moreover, the [110]-oriented Ge/Si core/shell NWs are expected to have a high mobility and a strong spin-orbit coupling [37,38] for the manipulation of hole spin qubits. Additionally, we believe this method is also applicable to obtain planar nanowires in other material systems with controllable size and orientation, such as III–V compound materials. However, the large V-groove poses a challenge for device fabrication, which needs to be addressed in future research work.

## Figures and Tables

**Figure 1 nanomaterials-11-00788-f001:**
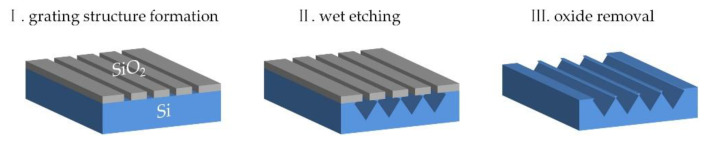
Schematic of process flow for the trapezoidal Si nanowire (NW) array.

**Figure 2 nanomaterials-11-00788-f002:**
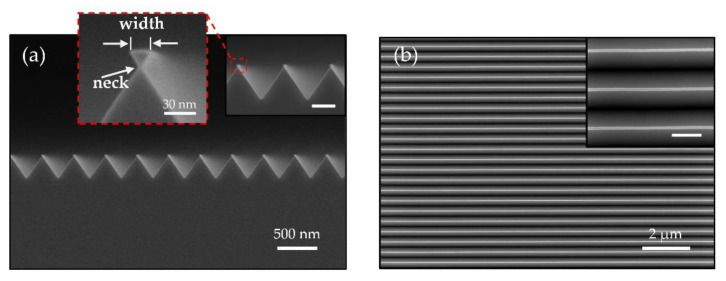
(**a**) Cross-sectional view and (**b**) top view SEM images of the Si NW array with an average wire width of 19 nm. Insets: zoom-in SEM images of the NWs. Scale bar of insets: 300 nm.

**Figure 3 nanomaterials-11-00788-f003:**
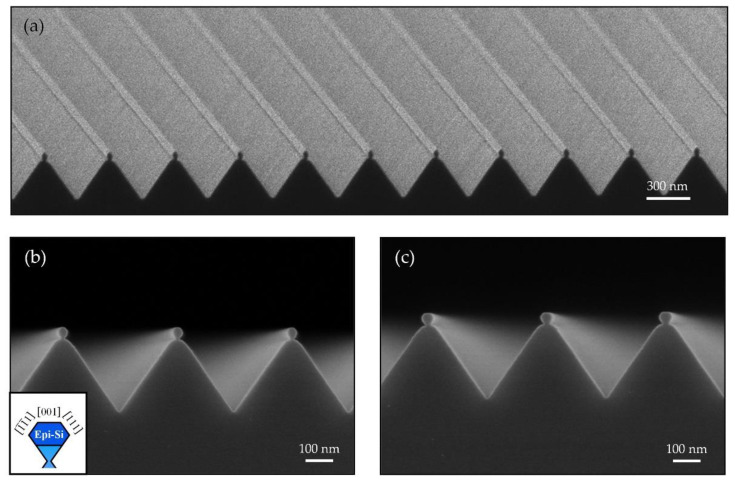
(**a**) Tilted SEM image showing the NW array of epitaxial Si on pre-patterned trapezoidal Si NWs. SEM images of epitaxial Si NWs obtained after the deposition of 20 nm Si at 380 °C on 30 nm wide pre-patterned trapezoidal NWs (**b**) and on 40 nm wide pre-patterned NWs at 380 °C (**c**). Inset of (**b**) schematically shows the truncated {111}-faceted cross-section.

**Figure 4 nanomaterials-11-00788-f004:**
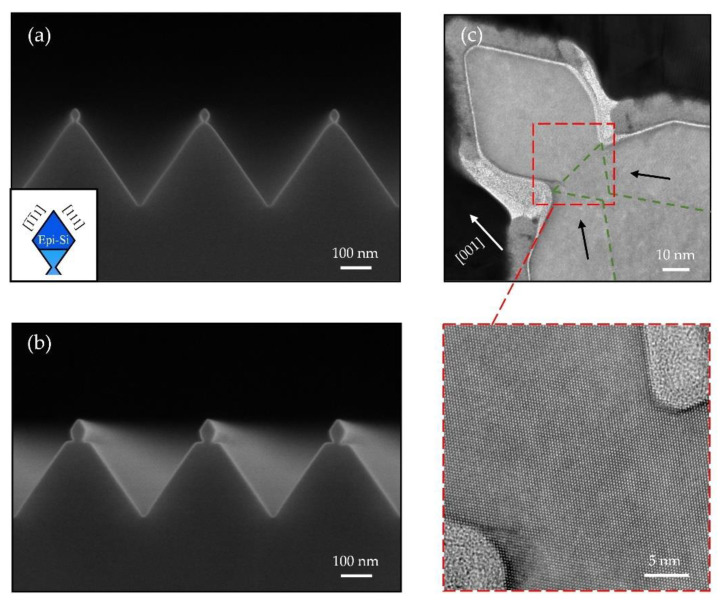
Cross-sectional SEM images of epitaxial Si NWs obtained after the deposition of 20 nm (**a**) and 50 nm (**b**) of Si at 450 °C on 30 nm wide pre-patterned trapezoidal NWs. Inset of (**a**) schematically shows the fully {111}-faceted cross-section. (**c**) Cross-sectional transmission electron microscopy (TEM) image of an epitaxial NW in (**b**), projected toward <110> direction. The interface of epitaxially formed Si NW and initial hourglass structure (pre-patterned Si NW) is sketched in green dashed line. The two shoulder areas marked in black are obtained by atomic diffusion during deposition. Inset of (**c**) shows a zoom-in high resolution transmission electron microscopy (HRTEM) confirming the perfect interface.

**Figure 5 nanomaterials-11-00788-f005:**
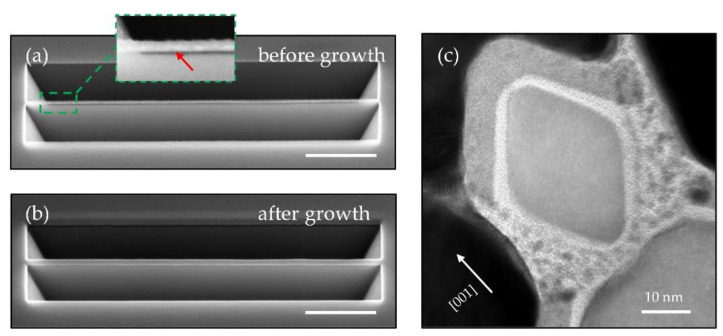
Suspended Si NW before (**a**) and after epitaxy (**b**). They both have a straight structure without distortion. Scale bar: 400 nm. (**c**) Cross-sectional HRTEM showing the high-quality diamond with four {111} facets after epitaxy. The red arrow in the inset of (**a**) highlights the suspended structure.

**Figure 6 nanomaterials-11-00788-f006:**
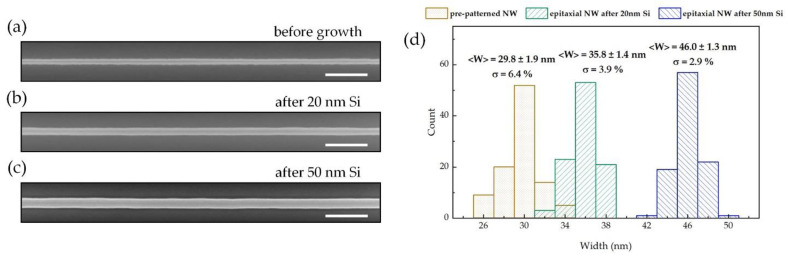
(**a–c**) SEM images of a 30 nm wide pre-patterned trapezoidal NW, epitaxial NWs after the deposition of 20 nm Si and 50 nm Si, respectively. Scale bar: 200 nm. (**d**) Statistical histogram showing the width distribution of 30 nm wide pre-patterned NWs, epitaxial NWs after the deposition of 20 nm and 50 nm Si layer. The average width <W> and relative standard deviation σ of the NWs are quoted.

**Figure 7 nanomaterials-11-00788-f007:**
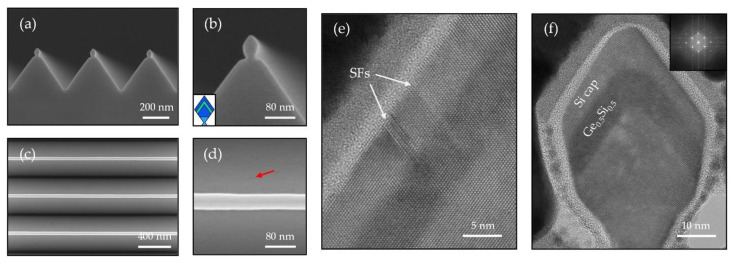
(**a**) Cross-sectional and (**c**) top view SEM images of the Si_0.5_Ge_0.5_ NW array and (**b**,**d**) the corresponding zoom-in images. The brighter contrast presenting in (**b**) results from the Si_0.5_Ge_0.5_ layer, where highlights in green in the schematic inset. The red arrow in (**d**) points to a strain-induced defect at the V-groove area. (**e**) Cross-sectional TEM image of SiGe at the V-groove area, showing that stacking faults (SFs) are generated from the interface and penetrate to the surface along the {111} gliding plane. (**f**) Cross-sectional HRTEM of a Si_0.5_Ge_0.5_ NW. Inset: FFT analysis of the SiGe/Si NW, showing a single set of spots indicating the SiGe is under fully strained condition.

**Figure 8 nanomaterials-11-00788-f008:**
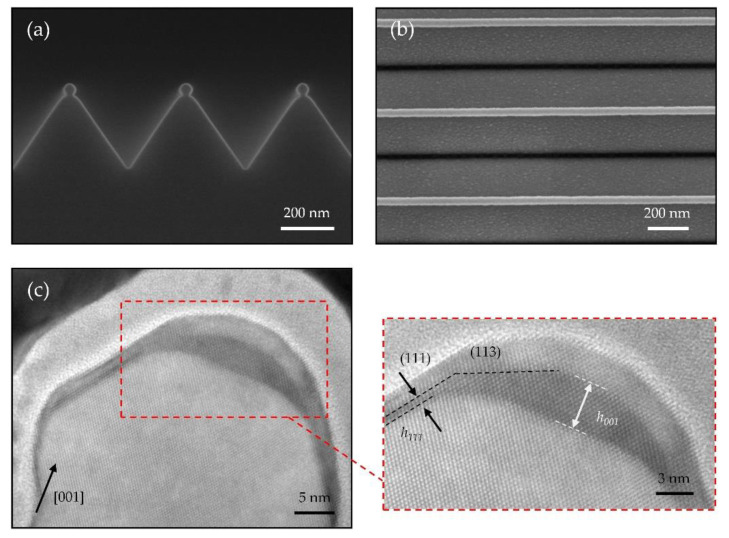
(**a**) Cross-sectional and (**b**) top view SEM images of the Ge/Si core/shell NW arrays. (**c**) Cross-sectional HRTEM image of a Ge/Si core/shell NW. Inset: a zoom-in HRTEM shows the two {113} side facets and the flat (001) top surface.

## Data Availability

Data available on request.

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
