# Peer review of "Epitaxial Growth of Ordered In-Plane Si and Ge Nanowires on Si (001)"

_nanomaterials, 2021, doi:10.3390/nano11030788_

Round 1
Reviewer 1 Report
The manuscript presents a detailed description of the wafer-scale molecular beam epitaxy growth of side-controlled in-plane Si a Ge nanowires. The NWs are deposited on pre-patterned 111-faced Si arrays. The authors clearly demonstrate that by adjusting the MBE growth parameters and/or the process of pre-patterning of the substrate they are able to control the morphology and dimensions of the NWs. The authors claim that the NWs are defect-free.
In general, the presented study concerns growth and morphological characterization of NWs. The presented results are not controversial. The manuscript reports new, interesting results; it is clearly written and does not contain false or misleading statements. I recommend publishing without changes
Author Response
Appreciate reviewer’s positive feedback.
Reviewer 2 Report
This manuscript describes the fabrication and HRTEM observations of in-plane Si, SiGe, and Ge nanowires on Si(001) substrate using molecular beam epitaxy. The authors have systematic HRTEM observations for nanowire shapes depending on constituent elements. The results of interest. However, there are several points that should be resolved.
- The authors discuss the results of Si, SiGe, and Ge/Si core/shell NWs in Sec. 3. However, in the title and the abstract there are few description for SiGe and Ge/Si core/shell NWs. They should be revised which are consistent with the results described in Sec. 3.
- In Sec. 2, for the fabrication of NWs, a simple figure which describes cartoons of fabrication processes is desirable.
- In the large inset of Fig. 1(a) scale bar should be provided.
- For Ge/Si core/shell NWs, it is not clear why (113) side facets are formed. Comments for physical origins for different face formation from Si NWs are desirable.
Author Response
Appreciate reviewer's suggestion.
- The authors discuss the results of Si, SiGe, and Ge/Si core/shell NWs in Sec. 3. However, in the title and the abstract there are few description for SiGe and Ge/Si core/shell NWs. They should be revised which are consistent with the results described in Sec. 3.
Thanks for reviewer’s advice, we add the description for SiGe and Ge/Si core/shell NWs in the abstract (line 13-14).
- In Sec. 2, for the fabrication of NWs, a simple figure which describes cartoons of fabrication processes is desirable.
According to reviewer’s suggestion, Fig. 1 is added into revised manuscript for describing the fabrication processes. (line 77-78)
- In the large inset of Fig. 1(a) scale bar should be provided.
The scale bars in the insets of Fig. 1 are included in the revised manuscript.
- For Ge/Si core/shell NWs, it is not clear why (113) side facets are formed. Comments for physical origins for different face formation from Si NWs are desirable.
We have provided the description of the physical origin as “The formation of Ge (113) facets is attributed to the low surface energy, which has been reported in previous works [31-33].” (line 250-251)
31Gai, Z.; Yang, W. S.; Sakurai, T.; Zhao, R.G. Heteroepitaxy of germanium on Si (103) and stable surfaces of germanium. Phys. Rev. B 1999, 59, 13009-13013.
32Gai, Z.; Ji, H.; Gao, B.; Zhao, R. G.; Yang, W. S. Surface structure of the (3× 1) and (3× 2) reconstructions of Ge (113). Phys. Rev. B 1996, 54, 8593.
33Laracuente, A.; Erwin, S. C.; Whitman, L. J. Structure of Ge (113): Origin and Stability of Surface Self-Interstitials. Phys. Rev. lett. 1998, 81, 5177.
Reviewer 3 Report
In this paper a method to fabricate by MBE Si and SiGe NWs arrays taking advantage of pre-patterned Si structures. Although not very novel, the nanostructures investigated here are of interest for the scientific community working on semiconductors.
The actual work done is relevant, with nice characterization results and extensive study of properties of these nanostructures. On the other hand, the quality of the manuscript should be improved, as the presentation of results lacks clarity in some parts and Introduction and Conclusions need consistent modifications.
In detail:
A) on page 2, lines 49-56: this section is more suitable in the conclusions - introduction should not be a long-form abstract. I'd recommend to move and summarize this part in the conclusions and replace it with a larger introduction on more recent results on Si and Ge NWs, in particular for control of direction of NWs. Authors should check also these other works on the topic:
1. Adhikari, H.; Marshall, A.F.; Chidsey, C.E.D.; McIntyre, P.C. Germanium nanowire epitaxy: Shape and orientation control. Nano Lett. 2006, 6, 318–323, doi:10.1021/nl052231f.
2. Constantinou, M.; Rigas, G.P.; Castro, F.A.; Stolojan, V.; Hoettges, K.F.; Hughes, M.P.; Adkins, E.; Korgel, B.A.; Shkunov, M. Simultaneous Tunable Selection and Self-Assembly of Si Nanowires from Heterogeneous Feedstock. ACS Nano 2016, 10, 4384–4394, doi:10.1021/acsnano.6b00005.
3. Fortuna, S.A.; Li, X. Metal-catalyzed semiconductor nanowires: A review on the control of growth directions. Semicond. Sci. Technol. 2010, 25, 024005, doi:10.1088/0268-1242/25/2/024005.
4. Toko, K.; Nakata, M.; Jevasuwan, W.; Fukata, N.; Suemasu, T. Vertically Aligned Ge Nanowires on Flexible Plastic Films Synthesized by (111)-Oriented Ge Seeded Vapor–Liquid–Solid Growth. ACS Appl. Mater. Interfaces 2015, 7, 18120–18124, doi:10.1021/acsami.5b05394.
5. Seravalli, L.; Bosi, M.; Beretta, S.; Rossi, F.; Bersani, D.; Musayeva, N.; Ferrari, C. Extra-long and taper-free germanium nanowires: use of an alternative Ge precursor for longer nanostructures. Nanotechnology 2019, 30, 415603, doi:10.1088/1361-6528/ab31cf.
B) page2, line 83 "50 Au" should be corrected. Did authors mean "50 nm Au"?
C) in HRTEM images of Fig.3 (c) and 4(c) what is the amorphous material around Si? Is it Si oxide? It seems to me the neck structure in Fig 3(c) and Fig 3(d) is asymmetric: is it due to the litographic process or does it appear after the growth? Could authors comment on this? Also, the green line in Fig.3(c) is hard to notice, could authors please make it more evident?
D) on pag.5 line 153: " The supporting Si necks are removed by similar fabrication method mentioned above with over-etched conditions." Could authors give more details on how the necks can be removed?
E) pag.5 line 154-155: this sentence is not very clear: what does it mean "to have superiorities"? Please rewrite it
F) end of page 5: is it possible to give the statistical error on the values of the NW width based on the reported values of standard deviation? It is not a common notation to give the values of standard deviation and leave to the reader to make the calculations. Please change text as: "widths of .... nm ± .... nm"
G) Figure 5 (d) is very obscure and difficult to understand due to very tiny font size. Please make it larger.
H) Very often, NWs are detached from the substrate with methods such as etching and/or sonication to have them in a solution. Could authors try to remove NWs from the substrate, drop-cast them on another inert substrate and make some statistical analysis about their dimensions (by SEM or similar techniques)? This should give informations about their resistance to detachment and their morphological properties as stand-alone nanostructures.
Some works to check:
Fabrication of Flexible and Vertical Silicon Nanowire Electronics - Jeffrey M. Weisse, Chi Hwan Lee, Dong Rip Kim, and Xiaolin Zheng, Nano Lett. 2012, 12, 6, 3339–3343 https://doi.org/10.1021/nl301659m
Wu, L., Li, S., He, W. et al. Automatic Release of Silicon Nanowire Arrays with a High Integrity for Flexible Electronic Devices. Sci Rep 4, 3940 (2014). https://doi.org/10.1038/srep03940
Orientation of germanium nanowires on germanium and silicon substrates for nanodevices - Beretta S Bosi M Seravalli L Musayeva NFerrari Materials Today: Proceedings, (2020), 30-36, 20 doi: 10.1016/j.matpr.2019.08.184
I) page 6, line 189: "That deposited" Please correct the mistake in grammar: it should read "That was deposited"
L) page 6, line 203: "the read arrow implies..." this verb is not correct here, please change it with "indicates" or similar words...
M) page7, line 239: "Considering the low growth temperature, the intermixing of Ge and Si is strongly suppressed, thus we can expect an almost pure Ge-core in such Ge/Si NWs." These are just speculations, please provide composition data (EDX or other composition-sensitive techniques) or remove this sentence. If authors do not provide actual data on compositions of these nanostructures, they should remove any reference to control of composition through the whole paper
N) Could authors comment on the maximum thickness possible to achieve with this method for Ge NWs? The thickness of Ge along the (111) direction seems to be very low (2-3 Monolayers?) What is the reason for such asymmetry? Also, did authors try to check the uniformity along the nanowire length? Is is possible to give an estimate of the changes in dimensions of the Ge NW if one changes zone of the nanowire?
O) Conclusions should be rewritten in full, they are too much emphatic: what does it mean to be "fully controllable"? I doubt "Super-homogeneous " is even a word. Conclusions should try to summarize the main findings of the paper, not trying to promote them. The fact that nanostructures are free of defects (not all them, actually) is demonstrated only by TEM, not by "various characterizations". Also, move here the part that I indicated above to remove from the Introduction.
Moreover, there is lack of perspective in the conclusions: I understand that this method allows to obtain NWs with a good control of size and orientation, but there are some points authors need to address:
1) what are strengths and weaknesses of this method in comparison with other approaches to have site-controlled NWs?
2) It seems to me that it is possible to obtain only in-plane NWs with this approach, not vertical ones: what are the disadvantages for this limitation in fabrication?
3) what are possible applications of these horizontal NWs? please make some specific, real-world examples citing real devices already developed and described in the literature. Vague and generic references will not be sufficient.
On the basis of these consideration, I'd recommend authors to revise this manuscript to improve it in order to deserve publication in NanoMaterials.
Round 2
Reviewer 3 Report
The authors have addressed in full all my concerns, so the paper it now suitable for publication in NanoMaterials